# Evaluation of Food Fineness by the Bionic Tongue Distributed Mechanical Testing Device

**DOI:** 10.3390/s18124250

**Published:** 2018-12-03

**Authors:** Jingjing Liu, Ying Cui, Yizhou Chen, Wei Wang, Yuanyuan Tang, Hong Men

**Affiliations:** 1College of Automation Engineering, Northeast Electric Power University, Jilin 132012, China; jingjing_liu@neepu.edu.cn (J.L.); 2201600406@neepu.edu.cn (Y.C.); w_wei110@163.com (W.W.); tyying1213@sina.com (Y.T.); 2Department of Computer Science and Bioimaging Research Center, University of Georgia, Athens, GA 30602, USA; 3Department of Neurobiology and Behavior, University of California, Irvine, Irvine, CA 92697, USA; yizhouc1@uci.edu

**Keywords:** texture perception, curvature difference, bionic tongue indenter, arrayed film pressure sensor, fineness perception evaluation model, sensory assessment

## Abstract

In this study, to obtain a texture perception that is closer to the human sense, we designed eight bionic tongue indenters based on the law of the physiology of mandibular movements and tongue movements features, set up a bionic tongue distributed mechanical testing device, performed in vitro simulations to obtain the distributed mechanical information over the tongue surface, and preliminarily constructed a food fineness perception evaluation model. By capturing a large number of tongue movements during chewing, we analyzed and simulated four representative tongue movement states including the tiled state, sunken state, raised state, and overturned state of the tongue. By analyzing curvature parameters and the Gauss curvature of the tongue surface, we selected the regional circle of interest. With that, eight bionic tongue indenters with different curvatures over the tongue surface were designed. Together with an arrayed film pressure sensor, we set up a bionic tongue distributed mechanical testing device, which was used to do contact pressure experiments on three kinds of cookies—WZ Cookie, ZL Cookie and JSL Cookie—with different fineness texture characteristics. Based on the distributed mechanical information perceived by the surface of the bionic tongue indenter, we established a food fineness perception evaluation model by defining three indicators, including gradient, stress change rate and areal density. The correlation between the sensory assessment and model result was analyzed. The results showed that the average values of correlation coefficients among the three kinds of food with the eight bionic tongue indenters reached 0.887, 0.865, and 0.870, respectively, that is, a significant correlation was achieved. The results illustrate that the food fineness perception evaluation model is effective, and the bionic tongue distributed mechanical testing device has a good practical significance for obtaining food texture mouthfeel information.

## 1. Introduction

The oral cavity is not merely an entry point of food, but also an important sensory region. Besides the taste and appearance, the texture is another important factor for consumer acceptability, that is much less well understood [1]. It is accepted that mechanical properties of food texture are the major contributors to our texture perception [2,3]. Although there are many mechanical sensory inputs in the oral cavity, such as lips and teeth, the tongue-palate compression is believed to play an important role [4]. In particular the distributed mechanical information provided by the tongue-palate compression has always been a hot spot in food texture research [5].

Oral cavity simulation is critical to the study of food texture perception. During the process of eating, the food undergoes constant changes inside the oral cavity, thus the oral cavity simulation is challenging [6]. Currently, research in this field includes simulating the oral cavity to study the eating physiology [7,8,9], exploring its internal transverse frictional force and radial shear force to develop an artificial saliva system, and designing a chewing evaluation model based on bionic teeth [10]. Chen et al. used “oral” tribology to study the food texture and the sensory perception, expounded the application prospects of tribology in the development and quality evaluation of food processing [11]. Sun et al. designed and developed a bionic food texture analyzer to simulate the mandibular chewing motion to detect the texture parameters such as hardness and brittleness [12,13]. Harrison et al. proposed an oral cavity force model based on smoothed particle hydrodynamics force to simulate the force interaction between the teeth and the tongue during eating [14]. From these simulation studies at home and abroad, it is apparent that exploring the food texture perception is a hot topic and these studies have made great contributions and are of practical significance to the exploration of food texture and sensory perception.

Our tongue perceives good tactile information through countless receptors located on its surface. When the tongue is in contact with food pieces, the receptors sense the distributed mechanical information over the flexible tongue surface [15,16,17,18], which plays a central role in obtaining the food texture perception [19]. Considering that the tongue movement in the oral cavity is a dynamic auxiliary chewing process, the surface distributed mechanical information on tongue surface changes overtime during eating, that is, with the change of the tongue state, the texture perception is a dynamic process [20]. However, there have been less studies on texture perception over different morphological tongue surfaces during chewing. Therefore, it is necessary to simulate some representative movement states of the tongue during chewing to promote the further development of food texture and sensory perception.

Currently, there are studies about food perception in terms of crispiness [21], springiness [22], viscosity [23,24,25], chewiness [26], but fineness perception is poorly studied or defined. Here, we defined “fineness” as a texture property with measurement of a degree of fine and grainy. For the study of food fineness, image analysis (IA) methods based on texture granulometry, electron microscopy, flatbed scanners and other instruments is usually used. Finally, the relationship between the sensory and the instrumental data is studied using multidimensional data analysis methods [27]. However, this paper intends to use the pressure data obtained from contact pressure tests to establish a food fineness perception evaluation model to quantify the perception of food fineness. Therefore, aiming at the dynamic changes of the tongue states during the auxiliary chewing process, we intend to simulate the tongue structure in vitro based on the differences of distributed curvature over the different morphologic tongue surfaces. Thus, the mechanical information involved in different morphological tongue surfaces will be collected, which can characterize the distributed mechanical information of food pieces over different morphological tongue surfaces during eating, and further explore the perception of food fineness. Here, we define the food fineness as the small uniformity degree of food pieces that are felt during the process of pressing the food, so that the real-time pressure data can be used to establish a food fineness evaluation model.

In this study, according to the law of the physiology of the mandibular movements and tongue movement features, we captured a large number of tongue movements during chewing. Considering that the dynamic process of exploring the tongue is not conducive to simulating the bionic model in vitro, therefore, we only analyzed and simulated four representative tongue movement states including the tiled state, sunken state, raised state, and overturned state of the tongue. By analyzing the curvature parameters and Gauss curvature of the tongue surface, we selected two circular regions of interest where the curvature distribution changed most dramatically in each tongue movement state, respectively. With that, eight bionic tongue indenters with different curvature over the tongue surface were designed. Together with an arrayed film pressure sensor, we set up a bionic tongue distributed mechanical testing device to design contact pressure experiments, which could simulate the process of the interaction between food and the tongue in different states, to detect the distributed mechanical information. According to the definition of fineness and the characteristics of experimental data, we have defined the fineness indicators by using mathematical ideas, which is beneficial for us to come up with a food fineness perception evaluation model based on the distributed mechanical information perceived by the surface of the bionic tongue indenters.

## 2. Construction of a Bionic Tongue Distributed Mechanical Testing Device

The tongue is the primary receptor in the mouth. By contacting with food pieces decomposed by the teeth and constantly adjusting the tongue position and shape, it can assist us in tasting, chewing and experiencing the perception of food texture. According to the law of the physiology of mandibular movements and the tongue movements features [28,29,30], the tongue motion during eating could be categorized into four major stages. First, the tongue is lying on the bottom of the oral cavity in a resting position, capturing a piece of food bitten by the frontal teeth and making full contact with food pieces. Secondly, the mouth closes and the oral space becomes smaller, while the central part of tongue is sunken to hold the food pieces. Third, the tongue rises to press the food against the maxilla bones to prepare for swallowing. Finally, the root of the tongue and epiglottis sense if the food pieces are small enough to be swallowed, if not, the tongue will turn out to push the food to be ground by the teeth, where it will be further broken. The above steps are repeated until the food pieces are suitable for being swallowed. Considering that the dynamic process of exploring the tongue is not conducive to simulating the bionic model in vitro, therefore, we only simulated four representative tongue movement states in the chewing process, including the tiled state, sunken state, raised state, and overturned state of the tongue [31]. These four representative tongue movement states were chosen to be the analysis templates, with which we obtained the surface tactile information of the tongue and further explored the food fineness parameter. Then we made contact pressure indenters named bionic tongue indenters in the shape of the tongue in these four states. Together with an arrayed type film pressure sensor, we finally constructed a bionic tongue distributed mechanical testing device, simulating the human tongue to obtain the food texture perception.

### 2.1. Bionic Tongue Indenter

To measure the distributed mechanical information of the tongue surface, it is necessary to consider constructing a bionic tongue indenter with the continuous tongue movement states. First, we should construct a 3D model of the four representative tongue states. Since the main difference of tongue surface forms is the curvature difference distributed over the surface, thus, the curvature parameter analysis and Gauss curvature analysis of the tongue surface will be done, then the bionic tongue indenter will be designed based on the above analysis.

#### 2.1.1. 3D Models Construction of Four Representative Tongue Movement States

First, considering the difference in the tongue shape between people, we selected 10 volunteers (five males and five females) and observed their four representative tongue movement states including the tiled state, sunken state, raised state, and overturned state of the tongue, respectively, during the chewing process. The tongue states were photographed by using a ZED binocular stereo camera. The multi-section curve of the tongue shape was drawn, and the normal specification of the multi-section section was obtained by using the ruler method to carry out the proper specification. The digitization models of four tongue states were produced by contour multi-section modeling and the guide lines. In the CATIA V5 R20 software, a 3D parametric modeling method was used to extract the surface data of the tiled, sunken, raised and overturned states of the tongue. The parameters recorded were the width of the tongue section *d*, the height of the tongue midpoint *h*, round at both ends *r_1_*, the round of the midline point *r_2_*. The mean value of each parameter is shown in Table 1.

Taking the overturned state of tongue as an example, the construction process of 3D model is shown in Figure 1. In the Generative Shape Design Module, first, we constructed a plurality of sections by using the plane splines and constrained the parameters such as the width of the tongue section *d*, the height of the tongue midpoint *h*, round at both ends *r_1_*, the round of the midline point *r_2_*. Secondly, according to the length of the tongue, the sections were sequentially translated by a certain distance in three-dimensional space, the purpose of which is to stretch the sections. The multi-section wireframe of the tongue was constructed by using the pose changes of the dotted frame (mostly translation and rotation). Then, according to the tongue contour parameter data and the symmetrical features of the tongue, we connected the contour lines and the center lines of the tongue sequentially by using the spatial spline. In the window of a multi-section surface definition, we completed the overall line structure for the multi-section by using closing points, guide lines and so on. Finally, after bridging the lines and smoothening the surface, we filled the model to a solid object, shown in Figure 2. Following the abovementioned steps, the 3D models of the other three tongue states were constructed too. The final 3D models of the four representative tongue movement states is shown in Figure 3.

#### 2.1.2. Statistical Analysis of Curvature Parameters

In the process of obtaining the mechanical information and simulating the food fineness perception over the tongue surface the main difference among the four tongue states (Tiled Tongue, Sunken Tongue, Raised Tongue, Overturned Tongue) lies in the curvature distribution on the tongue surface. Therefore, in this study, we chose to investigate the difference in food fineness perception by using statistical analysis of curvature parameters in the four different tongue states.

Due to the fact the curvature of a point characterizes the curve‘ bending condition of a small segment, therefore, in order to measure the difference in curvature distribution of the tongue surface, we only need to select some points that can fully reflect the curve curvature, and do not need to measure the curvature of each point on each curve [32]. According to the length of the tongue midline, the 3D tongue model was divided equally into eight segments, and correspondingly, seven sections were selected. According to the width of the tongue surface, the 3D tongue model was divided equally into four segments, and correspondingly, three sections were selected. Thus, the 3D tongue model was divided into seven coronal sections and three sagittal sections to analyze the curvature. The coronal section and sagittal section curve curvature radius were measured as L1-L7 and L(eft)/M(idle)/R(ight), respectively. 40 points and 80 points were selected from each coronal and sagittal section curve and the mean value was calculated based on every 10 such points as feature points. A non-parametric test (the Friedman test) was used to test the curvature difference among the points within each surface as *P_1_*, the sagittal sections curves as *P_2_*, the coronal sections curves as *P_3_*. Using the confident interval of 0.05. *P* value is the confidence that the difference among samples that are not from sampling error. The probability statistics are shown in Figure 4.

As seen in Figure 4, the *P_1_* value of Tiled Tongue is 0.028, which shows that it has a statistical difference between points. The *P_1_* value of Sunken Tongue is 0.000, which shows that it has an extremely significant statistical difference between points. The *P_1_* value of Raised Tongue is 0.005, which shows that it has a significant statistical difference between points. The *P_1_* value of Overturned Tongue is 0.334, showing no significant difference between points. It can be seen that four tongue states mainly differ in their curvature distribution characteristics, and this intrinsic property is the cause of the different tongue state. Different tongue states will also have different perceptions of the food texture. Therefore, the presence of the four tongue states is crucial.

As seen in Figure 4, we also analyzed the curvature of the points on three sagittal section contours. The *P_2_* values of Tiled Tongue, Sunken Tongue, Raised Tongue and Overturned Tongue are all greater than 0.05, showing that the curvature of the three sagittal sections curves all have no significant difference, indicating that the tongue midline curvature can represent well the other sagittal section curves’ curvature. In that the muscle in the midline of the tongue drives the tongue movement to assist chewing and breaking the food when people taste food, Table 2 shows the curvature radius of the feature points in the sagittal sections(L/M/R) in the four tongue states.

As for the coronal sections, the *P_3_* values of Tiled Tongue, Sunken Tongue, Raised Tongue and Overturned Tongue are all less than 0.05, showing that the curvature of the seven coronal section curves all have a significant difference, indicating that curvature of the tongue changes greatly in some local areas, causing a significant difference between the seven coronal sections curves. Table 3 shows the curvature radius of the feature points in the coronal sections (L1-L7) in the four tongue states.

In Table 2, it can be seen from the comparison of curvature differences about any two curves that there is no significant difference between the three sagittal sections curves (*P* > 0.05). In Table 3, after the comparison of curvature difference about any two coronal curves, firstly, we can see that two curves in Tiled Tongue between L1(31.76 ± 3.17) and L5(80.15 ± 21.54) have significant differences in curvature distribution (*P* < 0.05) and similarly, two curves in Tiled Tongue between L1(31.76 ± 3.17) and L6(209.41 ± 161.45) also have significant differences (*P* < 0.05). Second, two curves in Sunken Tongue between L3(80.57 ± 52.65) and L4(337.71 ± 110.35) has significant differences (*P* < 0.05). Thirdly, two curves in Raised Tongue between L1 (38.52 ± 23.53) and L3(76.15 ± 22.33) have significant differences (*P* < 0.05). Finally, two curves in Overturned Tongue between L2(51.19 ± 11.20) and L3 (165.78 ± 61.53) have significant differences (*P* < 0.05). From the curvature distribution of the coronal curves, we can know that the curvature distribution of different locations is not the same in each tongue position, and each has its own distribution characteristics.

As above, we analyzed the curvature difference in terms of the points of the curves of four tongue states. It can be observed that not only did the curvature differ significantly among states, but also the curvature distributions differ significantly within a state. The curvature of the local position over the tongue surface changes greatly, and some changes are small. The part where the curvature changes greatly is a representative area of the entire tongue surface, which can contain more useful information. Therefore, we will continue to further explore the curvature difference of the different local position within the same tongue surface to detect the representative regions with a large curvature change.

#### 2.1.3. Gauss Curvature Visualization Analysis

To further understand how the curvature changes within the same tongue state and detect the representative region(s) with a large curvature change, a Gauss curvature visualization analysis was conducted. Gauss curvature describes the bending degree of a curved surface around a point. By calculating the Gauss curvature of the discrete points on the tongue surface and combining with a Gauss curvature visualization map [33,34,35], we would be able to visualize the curvature distribution and select the areas that have the largest curvature changes. Such areas can preserve most of the curvature information of the tongue surface.

We introduced a Voronoi algorithm based on the discrete Gauss curvature analysis [36]. The algorithm treats the smooth curve as the limit or linear approximation of a family of grids. The metric property of each vertex in the triangular mesh is regarded as an average measure of a small neighborhood at this point. By calculating its mixture area aiming at each point, Gauss curvature of the point was obtained through the following formula. The Gauss curvature was discretized by using the Gauss-Bonnet theorem in classical differential geometry. The current discrete Gauss curvature was expressed as the Gauss curvature at point P in the irregular tongue surface:(1)$$\overset{1}{\underset{Am}{\iint }}KdA=2\pi -\underset{j}{\sum }\theta_{j}$$ where $\theta_{j}$ is the angle between edges $v_{i}v_{j}$ and $v_{i}v_{j+1}$. Discretizing the integral we obtain the discrete form of the Gauss curvature, as shown in the following formula:(2)$$K_{G}\left(P_{i}\right)=\frac{1}{A_{m}}\left(2\pi -\underset{j\in N\left(i\right)}{\sum }\theta_{j}\right)$$ where $K_{G}$ is Gauss curvature. $P_{i}$ is the vertex of the triangle mesh. N(i) is the subscript range within its 1-field. $A_{m}$ are the mixing areas of the triangular meshes around the point P. The Gauss curvature array was normalized to [−1,1]. The normalization formula is shown in the following expression:(3)$$K_{G}^{\prime }=\frac{K_{G}-K_{G_{min}}}{K_{G_{max}}-K_{G_{min}}}$$ where $K_{G}^{\prime }$ is the Gauss curvature after normalization. $K_{G_{max}}$ is the maximum value in the array. $K_{G_{min}}$ is the minimum value in the array. According to the Gauss curvature symbol we can use the absolute value size to describe the direction of bending and by using a color gradient and the gradient mode of an equivalent cloud map for pseudo-color display on the tongue surface, the Gauss curvature visualization map was finally obtained, as shown in Figure 5. 

Figure 5 shows the intuitive Gauss curvature visualization map and curvature difference distribution for the Tiled Tongue, Sunken Tongue, Raised Tongue and Overturned Tongue. First, the Gauss curvature value of each point in the tongue surface was calculated. Then, we illustrate the curvature difference by using a color code, with red representing the largest curvature and blue representing the lowest curvature. The curvature change rate could also be seen through the gradual change of colors.

From the curvature distribution map above, we proceeded to find regional circles with a large change in color on the tongue surface that could represent a large part of the tongue surface. Representative regions could be used instead of the whole tongue surface to construct the bionic tongue indenter, because they hold most of the tongue surface curvature information, that is, sense a large amount of food texture. From the Friedman test results, which is shown in Figure 4, we know that the curvature did not differ significantly along the sagittal section curves, which is also supported by the Gauss curvature distribution map. Thus, we plotted the curvature change along the tongue midline corresponding to the position of coronal sections curves (L7–L1), which shows that the curvature distribution of the tongue midline on each tongue surface has multiple peaks and valleys. Moreover, Gauss curvature visualization maps show that they are change relatively faster from blue to red or from red to blue in the front tongue and end tongue. Thus, we selected two regional circles of interest where the curvature distribution changes the most dramatically within each tongue state, respectively. Figure 6 shows the origin points of the regional circle of interest in the tongue midline for the four tongue states.

According to the position of the origin points, we selected two regional circles of interest having a diameter of 20 mm in each tongue surface, marking them as Appendix A, as shown in Figure 5. As can be seen in Figure 5, more curvature changes are shown within the regional circle of interest, which can represent to sense a large amount of food texture information. As shown in Figure 5a, Appendix A are centered at a distance of 25 mm between the horizontal planes. As shown in Figure 5b, Appendix A are tangent to each other. As shown in Figure 5c, Appendix A are centered at a distance of 27 mm between the horizontal planes. As shown in Figure 5d, Appendix A are tangent to each other.

#### 2.1.4. Making the Bionic Tongue Indenter

On the CATIA V5 R20 platform, the tongue in the circle of the abovementioned area of interest is obtained by using the groove tool of the part design module, and the morphological structure of the upper surface of the tongue is preserved. To be convenient to install, the protrusion tool was used to add a quadrilateral prism with the height of 20 mm and the length of side of 10 mm. After saving the 3D virtual model as a STL. format file, a 3D printer was used to print the bionic tongue indenter model. The filling rate was 80% and PLA material was used. This material has high mechanical strength, pressure resistance and heat resistance. It also has a certain resistance to bacteria, which is suitable for contacting food. Finally, eight bionic tongue indenter solid models were obtained, which were respectively named as the front of Tiled Tongue S1, the end of Tiled Tongue S2, the front of Sunken Tongue S3, the end of Sunken Tongue S4, the front of Raised Tongue S5, the end of Raised Tongue S6, the front of Overturned Tongue S7 and the end of Overturned Tongue S8, as shown in Figure 7. 

### 2.2. Arrayed Film Pressure Sensor

The tongue makes contact with the food directly during eating process and senses the distributed mechanical force on the tongue surface, which is crucial for the texture perception. In order to simulate the tongue surface perception, it is necessary to combine the indenters with a tactile sensor with good flexibility to be capable of multi-point density measurement. For this reason, an arrayed film pressure sensor has been used in this work to perform array layout of the sensing units for the eight bionic tongue indenters made by simulating the tongue structure, so as to realize the collection of mechanical force information at the multi-pressure sensing points over the bionic tongue indenter surface.

The arrayed film pressure sensor used in this work is a resistive pressure sensor, which has an “interlayer” structure, composed of two layers of films, a conductive electrode and a pressure-sensitive semiconductor material coating. The inner surfaces of both polyester films have transverse strip conductors or longitudinal strip conductors. A pressure-sensitive semiconductor material is sandwiched between two films. When the arrayed sensor is pressed, the resistance of the pressure sensitive semiconductor changes with the proportion to the external force, and the intersection of the horizontal conductors and the longitudinal conductors forms an array of pressure sensing points (PSP), whose sensing matrix is 52 × 44. The line interval of the sensor is 1 mm and the minimum area of the resolution unit is 2 mm×2 mm. That is, the rank of the sensing area of this sensor is 17 × 12.8 cm^2^. The sensor is packaged by a PET film with a thickness of 100 μm. PET film is non-toxic, chemically-resistant, friction-resistant, low-wear and high-hardness, and has greatest toughness among thermoplastics, which is good for us to do food experiments. What’s more, for this arrayed film pressure sensor, the sealing thickness is about 0.2 mm, the measuring range is 0–50 kg and the precision is ≤±20%. The structural diagram of the arrayed film pressure sensor is shown in Figure 8.

### 2.3. Construction of Distributed Mechanical Testing Device

The bionic tongue distributed mechanical testing device also has a lifting device with an attachment, a handle for data acquisition in addition to the bionic tongue indenter and the arrayed film pressure sensor. Figure 9 shows the bionic tongue distributed mechanical testing device.

For this device, we selected a Handpi Force Gauge as the lifting device, including the support guide pillar, pressure arm and pedestal. Firstly, the bionic tongue indenters used in the experiment are installed on the fixture in the lifting device. The pressure arm is controlled to decline along the guide pillar. Thus, the bionic tongue indenter is contacted with the sample on the pedestal and continuously imposes external force on the samples. When it is compressed to some extent, th pressure arm will automatically go back to the initial position. The stroke range is 0–580 mm and the testing speed is 0.5–5 mm/s. The maximal load of Handpi Force Gauge is 500 N. The speed button on the pedestal is adjusted to set the automatic lifting speed of the pressure arm. What’s more, the times of cyclic compression testing can be pre-established, which can be realized on the electronic screen of the pedestal. 

The bionic tongue indenter, the arrayed film pressure sensor, the handle for data acquisition and the lifting device are constructed in the order of assembly to complete the bionic tongue distributed mechanical testing device. First, the bionic tongue indenter needs to be installed at the end of the pressure arm in the experiment. Then, the arrayed film pressure sensor needs to be placed on the steel plate. The position is adjusted to locate the central area below the bionic tongue indenter. The data acquisition side of the handle is connected to the arrayed film pressure sensor, and the other side is connected to the computer through a USB port. Thus, the construction of the bionic tongue distributed mechanical testing device has been completed. 

## 3. Experimental Section

### 3.1. Experimental Material

To ensure other texture attributes are consistent and perform a single fineness analysis, in this paper, different kinds of cookies were selected as the experimental material. According to the classification methods of mechanical properties concerning food texture, we referred to GB/T 29604-2013 Sensory Analysis-General Guidelines for Establishing a Reference for Sensory Characteristics [37] and selected three kinds of cookies with differences in fineness texture characteristics, identified as WZ Cookie, ZL Cookie and JSL Cookie. For the WZ cookie samples with irregular shapes, we used the smallest unit to measure. For the ZL Cookie, and JSL Cookie samples, with regular shapes, we made the cookie into a cube of 1 cm^3^. We take cookies without cracks as the experimental samples. After each test, the bionic tongue indenter and the stage were cleaned to ensure the consistency of measurement conditions.

### 3.2. Experimental Method

The analysis of the mechanical properties of food is an important factor affecting the quality of the food, directly affecting the sensory quality and consumer acceptance of the food. Therefore, it is proposed to analyze the texture properties of food by a mechanical pressure test. Before the experiment, a bionic tongue indenter was installed on the fixture in the lifting device. Then, a sample to be measured was placed on the arrayed film pressure sensor, whose position was adjusted to place it below the bionic tongue indenter. When the sample was compressed to some extent, the arm would automatically go back to the initial position. Next, the analysis software conforming to the arrayed film pressure sensor was opened and the sampling frequency was set 33 Hz. Afterwards, we set the number of times of cyclic compression testing of the pressure arm to one time with a speed of 2 mm/s, and the lowest point from the pedestal was set to 2 mm. Lastly, the machine table was started, followed by using the eight bionic tongue indenters to perform the contact pressure tests. Each test required placement of a new sample. The pressing speed and the compression degree of the bionic tongue indenter remained unchanged.

### 3.3. Sensory Assessment

The food sensory assessment is a kind of statistical-based psychological activity, which is a quantitative and qualitative description of the food by consumers through sensory perception. According to the national standard GB/T 10220-2012 Sensory analysis—Methodology—General guidance [38]. 20 evaluators (10 men and 10 women) were selected and trained to constitute an evaluation group to conduct sensory assessment on the sample texture of WZ Cookie, ZL Cookie and JSL Cookie. Under the precondition of guaranteeing sample texture features and basic sampling principles, the samples were numbered and submitted to the evaluator for independent evaluation according to a random coding. What’s more, to avoid subjective psychological cues due to its appearance and color we should ensure the evaluators wear an eye mask. The mouthwash was used to remove any residual taste in the oral cavity between two samples or two evaluation activities. Moreover, we used a linear scale detection method to mark the intensity of the sensory perception on a linear scale of 10 cm. The two endpoints of the 10 cm line correspond to the lowest and highest score of the sensory assessment. Each sample was evaluated three times and the result summarized by using the mean value method. In the end, the linear scale was converted into the corresponding proportional value for statistical analysis.

## 4. Results and Analysis

### 4.1. Contact Pressure Distribution

In this experiment, there are three kinds of cookies, identified as WZ Cookie, ZL Cookie and JSL Cookie. Each kind of cookie has three parallel samples. The device in Figure 9 was used for the contact pressure tests. According to the above experimental method, each parallel sample was tested under the eight bionic tongue indenters. In the data collection interface of the analysis software, the image of pressure distribution following time in the sensing region during the constant velocity advancement of the arrayed film pressure sensor is displayed in real time. The image was featured by the annular distribution with the center of the peak stress. The distributed mechanical image and the matrix of pressure values corresponding to the image at each moment has been stored in the data frame. We set the device to acquire one data frame every 0.02 s in this experiment.

The data frame covered from contacting the three kinds of cookies to leaving the pieces completely is different. According to the characteristics of the experimental data among the three kinds of cookies, the maximum range in the data frame had been selected as the subsequent calculated sample size, that is, the 18–138 data frames had been defined as the valid data frame. Within the valid data frame, the maximum planar matrix of pressure value had been 13 × 11. The middle sensing area 4.2 × 3.6 cm^2^ of the arrayed film pressure sensor was defined as the valid pressure area. 

Considering that the curvature distribution of the eight bionic tongue indenters’ surface has a large difference, which results in the mechanical information perceived by the different bionic tongue indenters being different. The images corresponding to the maximum total pressure in the JSL Cookie sample as perceived by the eight bionic tongue indenters are shown in Figure 10.

From Figure 10, it can be seen that the bionic tongue distributed mechanical testing device can obtain a large amount of experimental data. The pressure matrix corresponding to the image over the surface in each data frame during the contact pressure process can lay the foundation for the study of food fineness.

### 4.2. Fineness Perception Evaluation Model Establishment and Solution

In this study, we used the distributed pressure data from the valid data frame of the contact pressure distribution map to establish a food fineness perception evaluation model. We define “fineness” as a texture property with measurement of fine and grainy characteristics. In this paper, the data frame at each moment stores the pressure value of the food pieces scattered on the arrayed film pressure sensor during the process of pressing the food. Here, we define the food fineness as the small uniformity degree of the food pieces that are felt during the process of pressing the food, so that the real-time pressure data can be used to establish a food fineness evaluation model. Since the finer the density of the food, the more uniform the force distribution felt by the pressure sensing sites on the tongue surface, the more uniform the force distribution felt by the PSP in adjacent data frames. Therefore, according to the pressure distribution change data, we selected three major indicators to establish the food fineness perception evaluation model.

#### 4.2.1. Gradient of Bionic Tongue PSP G(i,j)

The stress value of the sensing area on the bionic tongue surface is a scalar field. In vector calculus, the gradient length of a certain point in the scalar field indicates the rate of change in that direction. Calculate the values of the horizontal gradient and the longitudinal gradient, the difference in the gradient values between the maximum value and the minimum value in each PSP in each frame is defined as a gradient index value G(i,j). Let the mechanical information sensing area be S, the gradient value of the PSP(x,y) along the *x*-axis direction in the sensing area is indicated in the following formula:(4)$$G_{s,x}=\frac{\partial G}{\partial x}$$

Gradient values along the *y*-axis direction are indicated in the following formula:(5)$$G_{s,y}=\frac{\partial G}{\partial y}$$

The process of calculating the gradient index value is introduced:(6)$$O\left(j,:\right)=\frac{\left[\left(G_{s,x}\left(j,:\right)-G_{s,x}\left(j-1,:\right)\right)+\left(G_{s,x}\left(j+1,:\right)-G_{s,x}\left(j,:\right)\right)\right]}{2},\;\;\;\;1<j<m$$
(7)$$P\left(:,i\right)=\frac{\left[\left(G_{s,y}\left(:,i\right)-G_{s,y}\left(:,i-1\right)\right)+\left(G_{s,y}\left(:,i+1\right)-G_{s,y}\left(:,i\right)\right)\right]}{2},\;\;\;\;\;\;1<i<n$$
(8)$$A_{i,j}=max\left[O\left(j,:\right),P\left(:,i\right)\right],\;B_{i,j}=min\left[O\left(j,:\right),P\left(:,i\right)\right],\;\;\;\;\;\;1<j<m,\;1<j<n$$
(9)$$G_{i,j}=A_{i,j}-B_{i,j}$$ where $G_{i,j}$ is the gradient index value of one point. O(j, :) is the horizontal gradient absolute value of the matrix. P(:,i) is the longitudinal gradient absolute value of the matrix. $A_{i,j}$ is the maximal value among the horizontal gradient value and longitudinal gradient value in the matrix. $B_{i,j}$ is minimum value among the horizontal gradient value and longitudinal gradient value in the matrix.

#### 4.2.2. Stress Change Rate of Bionic Tongue PSP $M_{i,j}$

The stress change rate represents the change of the stress value per unit of time. The stress value of each PSP is changing over time. The magnitude of the stress on the food pieces changes constantly, which reflects that the stress value of each PSP changes constantly in each data frame. We calculate the stress values’ difference of a certain PSP between the previous frame and the following frame, expressed as $\Delta M_{T,ij}$. The difference between the maximum value and the minimum value in each difference is defined as the polarity of a PSP, expressed as $M_{i,j}$:(10)$$\Delta M_{T,ij}=M_{T,ij}-M_{T-1,ij\;\;\;\;\;\;\;}1<j<m,\;1<j<n$$

(11)$$M_{i,j}=\max\left(\frac{\Delta M_{T,ij}}{\Delta t}\right)-min(\frac{\Delta M_{T,ij}}{\Delta t})\;\;\;\;\;\;1<j<m,\;1<j<n$$

#### 4.2.3. Areal Density of Bionic Tongue PSP $\rho_{sz}$

In geometry, the areal density represents the quality per unit area of a substance with a certain thickness, it can characterize the fine degree of a food mass scattered at each PSP. The total mass to area ratio of the food pieces in contact with each PSP is defined as an areal density index value $\rho_{sz}$:(12)$$\rho_{sz}=\frac{\sum_{z=1}^{n}G_{z}}{\sum_{z=1}^{n}S_{z}}$$ where $G_{z}$ is the quality of the food pieces in contact with one PSP *z*. $S_{z}$ is the area of the food pieces in contact with one PSP *z*.

#### 4.2.4. Establishment of Fineness Perception Evaluation Model

The above three indicators including gradient, stress change rate and areal density are all the indexes used to describe the fineness, which are weighted to get the comprehensive fineness index *I*:(13)$$I=\alpha G_{min}+\beta M_{min}+\gamma \rho_{min}$$ where *α*, *β*, *γ* are the weight coefficients of gradient value, stress change rate, and areal density, respectively. 

To avoid the influence of subjective randomness, we used the entropy weight method to determine the weight objectively for the three indicators including gradient, stress change rate, and areal density. The entropy weight method uses the information entropy to calculate the entropy of each index based on the degree of variation. Then, the weight of each index is corrected by the entropy weight, and the objective weights of indicators are obtained. The procedure is follows:
(1)Data Standardization

The matrix $I={\left(i_{uv}\right)}_{st}$, $i_{uv}$ represents the value corresponding to the *v*-th index of the *u*-th evaluation object (*u* = 1, 2, ⋯, 72, corresponding to the three kinds of cookie-Cookie, ZL Cookie, JSL Cookie—under eight bionic tongue indenters; three samples are measured for each kind of cookie under each bionic tongue indenter, *v* = 1, 2, 3, corresponding to three indicators including gradient value, stress change rate and areal density). The matrix is represented as:(14)$$I={\left[\begin{array}{ccc} I_{11} & I_{12} & I_{13}\\ I_{21} & I_{22} & I_{23}\\ \cdots & \cdots & \cdots \\ I_{u1} & I_{u2} & I_{u3} \end{array}\right]}_{72\times 3}$$

The standardized formula is represented as:(15)$$l_{uv}=\frac{i_{uv}-min\left(i_{u.}\right)}{max\left(i_{u.}\right)-min\left(i_{u.}\right)}$$

The standardization matrix $L={\left(l_{uv}\right)}_{st}$ is obtained by the following calculation:(16)$$L={\left[\begin{array}{ccc} L_{11} & L_{12} & L_{13}\\ L_{21} & L_{22} & L_{23}\\ \cdots & \cdots & \cdots \\ L_{u1} & L_{u2} & L_{u3} \end{array}\right]}_{72\times 3}$$


(2)Index Information Entropy Acquisition


The information entropy *Ev* of the *v*-th indicator can be expressed as:(17)$$E_{v}=-\frac{1}{\ln\left(s\right)}\overset{s}{\underset{u=1}{\sum }}p_{uv}\ln\left(p_{uv}\right)$$ where, $p_{uv}=l_{uv}/\sum_{u=1}^{s}l_{uv}$, when $p_{uv}$ = 0, then the following equation is established as:(18)$$\lim\left(p_{uv}lnp_{uv}\right)=0$$

If the entropy value *E_v_* of a certain index is smaller, illustrating that the degree of variation of the index value is greater, the more amount of information it provides, the greater the role of the index in comprehensive evaluation, and the greater the weight should be.

(3)Weight of Each Indicator Acquisition

The weight expression of the *v*-th indicator is as follows:(19)$$w_{v}=\frac{1-E_{v}}{\sum_{v=1}^{t}\left(1-E_{v}\right)}$$

The result of the calculation was expressed as: *w*_1_ = 0.181; *w*_2_ = 0.517; *w*_3_ = 0.303, that is, the weight coefficients of three indicators including gradient, stress change rate, areal density were equal to 0.181, 0.517, 0.303.

The indicator weight values obtained were brought into the model Equation (13). According to the food fineness perception evaluation model established based on the sensing mechanical information on the bionic tongue indenter surface, the fineness index values under the different bionic tongue indenters and the different kinds of cookie samples were calculated. Figure 11 shows the average of the fineness index values of the three kinds of cookies in steady status under the eight bionic tongue indenters.

According to the fineness index values obtained from the model, the fineness index values of the three kinds of cookies were plotted to fit the stress-fineness curve under the eight different indenters. Figure 12 shows only the fitted stress-fineness curve of the three kinds of cookies under the bionic tongue indenter S3.

Figure 12 shows the changing trend of the fineness index value, which is following the gradually increasing pressure on WZ Cookie, ZL Cookie and JSL Cookie by using the S3 bionic tongue indenter. According to the three defined indicators, the smaller the gradient index value is, the finer the food pieces in the frame are. The smaller the stress change rate, the finer the PSP over the tongue surface perceived. The smaller the areal density, the larger the contact area, indicating that the food piece is finer. Therefore, the smaller the fineness index value is, which indicates that the food pieces are finer. From the trend of the three curves, it can be seen that as the stress on the cookie block increases, the overall trend of the stress-fineness fitting curve is gradually decreasing and tending to be stable, indicating that following the increasing pressure on the food block, the tongue surface can perceive that food pieces are becoming more and more fine, and the fineness index eventually tends to a stable value. In the curve changing trend, it can be seen that the fineness of JSL Cookie is poor, and the fineness of WZ Cookie and ZL Cookie are similar. However, from the overall trend of curves, it can be seen that the fineness of WZ Cookie is better, and from the value of the index when it finally stabilizes, it also can be known that the fineness index value of WZ Cookie is smaller. Similarly, under the other indenters, the stress-fineness fitting curves of the three kinds of cookies show that the WZ Cookie has the best fineness, followed by ZL Cookie, and the fineness of the JSL Cookie is the worst, which is in line with the actual fineness perception of the human tongue.

### 4.3. Sensory Assessment

The WZ Cookie, ZL Cookie, and JSL Cookie were evaluated by 20 evaluators for sensory assessment. The results were recorded on a straight line of 10 cm in length. We measured it and performed data statistics, among them, the two endpoints of the 10 cm line were respectively corresponding to the lowest score and the highest score of the sensory assessment, that is, the higher the score, the finer the perception. The fineness perception score results of the three kinds of cookies are as shown in Figure 13.

Figure 13 shows the sensory scores of the three kinds of cookies by 20 evaluators. It can be seen that the fineness sensory scores of WZ Cookie are the highest. JSL Cookie has a lower score. It shows that the true fineness perception of humanity is that the fineness of WZ Cookie is the best, followed by the ZL Cookie, and the fineness of JSL Cookie is worst. The conclusion of the sensory assessment and the results from above-mentioned experiment are the same.

### 4.4. Correlation Analysis between Sensory Assessment and Fineness Evaluation Model Results

In order to explore the correlation between the food fineness perception evaluation model results established based on the mechanical information perceived by the bionic tongue indenter and the sensory assessment. Pearson correlation analysis was used to calculate the correlation coefficient and to obtain the correlation coefficient matrix. The correlation coefficient between the two variables *X*, *Y* is defined as:(20)$$\rho_{XY}=\frac{Cov\left(X,Y\right)}{\sqrt{Var\left(X\right)}\sqrt{Var\left(Y\right)}}=\frac{E\left[\left(X-EX\right)\left(Y-EY\right)\right]}{\sqrt{E{\left(X-EX\right)}^{2}}\sqrt{E{\left(Y-EY\right)}^{2}}}$$ where *X* is the sensory score of food fineness. *Y* is the fineness index calculated by the evaluation model. Cov(X,Y) is the covariance of variable *X* and *Y*. *EX* is the mean of *X*. *EY* is the mean of *Y*. *Var*(*X*) is variance of *X*. *Var*(*Y*) is the variance of *Y*.

The sample correlation coefficient is expressed as:(21)$$r=\frac{\sum_{i=1}^{n}\left(x_{i}-\bar{x}\right)\left(y_{i}-\bar{y}\right)}{\sum_{i=1}^{n}\left(x_{i}-\bar{x}\right)\sum_{i=1}^{n}{\left(y_{i}-\bar{y}\right)}^{2}}$$ where $x_{i}$ is a random sample of capacity *n* taken from variable *X*. $y_{i}$ is a random sample of capacity *n* taken from variable *Y*.

Table 4 shows that we obtained the correlation coefficients of the three kinds of cookies between the sensory assessment and fineness index values under the eight bionic tongue indenters, and the average value of the WZ Cookie’s correlation coefficients is 0.887, the average value of the ZL Cookie’s correlation coefficients is 0.865, the average values of the JSL cookie’s correlation coefficients is 0.870, which are significantly correlated. Among them, four bionic tongue indenters including the front of Tiled Tongue S1, the end of Raised Tongue S6, the front of Overturned Tongue S7 and the end of Overturned Tongue S8 have the fineness sense of WZ Cookie, JSL Cookie and the fineness perception results have extremely significant correlations with the sensory assessment. This shows that the food fineness perception evaluation model is effective, and the bionic tongue distributed mechanical testing device has good practical significance for simulating the food texture perceptions.

## 5. Discussion

In this article, according to the law of the physiology of mandibular movements and tongue movements features, we captured a large number of tongue movement states during chewing, analyzed and simulated four representative tongue movement states, including Tiled Tongue, Sunken Tongue, Raised Tongue, and Overturned Tongue. In addition, by analyzing the curvature parameter and Gauss curvature of the tongue surface, we selected the regional circle of interest where the curvature distribution changed the most dramatically. The middle point of the tongue midline difference between the maximal peak and minimal valley was selected as the origin point of the regional circle of interest. The body of tongue within the regional circle of interest was 3D printed as eight bionic tongue indenters with different curvatures. In combination with an arrayed film pressure sensor capable of multi-point density measurements, a bionic tongue distributed mechanical testing device was built to perform in vitro simulations and obtain the distributed mechanical information over the tongue surface.

We performed a contact pressure experiment on three kinds of cookies—WZ Cookie, ZL Cookie and JSL Cookie—with different fineness texture characteristics. A food fineness perception evaluation model was built by using the pressure values in the valid data frame. The model was achieved by defining three indicators including gradient, stress change rate, areal density and determining the weights. From the stress-fineness curves obtained by the model, it can be seen that the stress on the cookie block increase, the fineness index gradually decreases and tends to be stable. The fineness of WZ Cookie is the best, followed by ZL Cookie, and the fineness of the JSL Cookie is the worst, which is in line with the actual fineness perception of the human tongue.

The correlation between the fineness index values calculated by the model and sensory assessment was analyzed. The results show that the average of the correlation coefficients under the eight bionic tongue indenters reached 0.887, 0.865, and 0.870, respectively, that is, a significant correlation. The results illustrate that the food fineness perception evaluation model is effective, and the bionic tongue distributed mechanical testing device results are reasonable.

## 6. Conclusions

In this study, the bionic tongue distributed mechanical testing device constructed to simulate tongue perception can obtain a large number of distributed pressure values. The food fineness perception evaluation model was established by using the large amount of data, which can provide practical significance for exploring the tactile perception of the tongue in the oral environment. The research in this paper has the potential application values in oral processing, oral physiological characteristics, physiological dietary habits, which are crucial for the formation of the food texture perception. The food industry urgently needs further research to better explore the texture parameters of food.

## Figures and Tables

**Figure 1 sensors-18-04250-f001:**
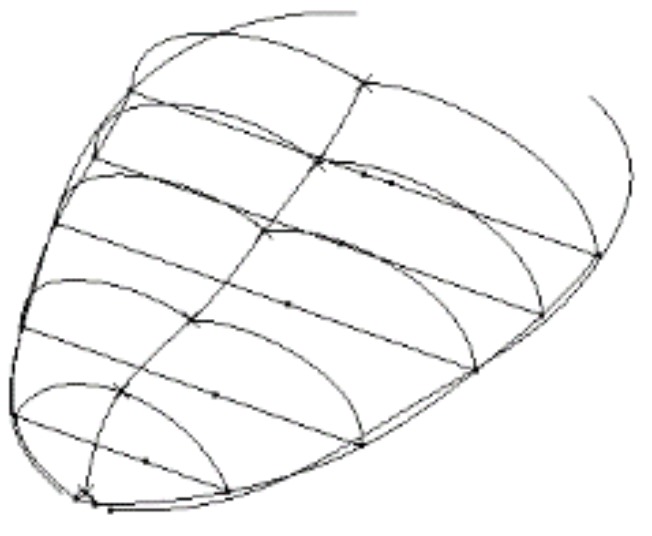
Overall line architecture of tongue.

**Figure 2 sensors-18-04250-f002:**
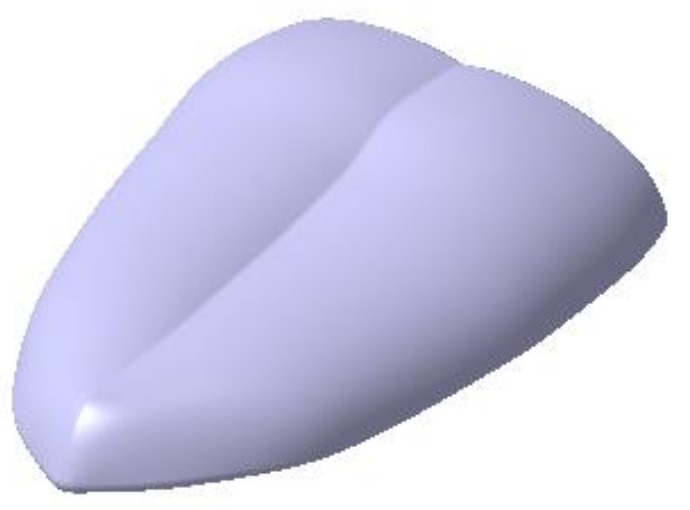
3D model of the tongue.

**Figure 3 sensors-18-04250-f003:**
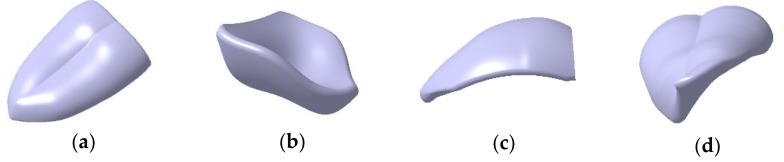
3D model of four representative tongue movement states: (**a**) Tiled State of Tongue (Tiled Tongue); (**b**) Sunken State of Tongue (Sunken Tongue); (**c**) Raised State of Tongue (Raised Tongue); (**d**) Overturned State of Tongue (Overturned Tongue).

**Figure 4 sensors-18-04250-f004:**
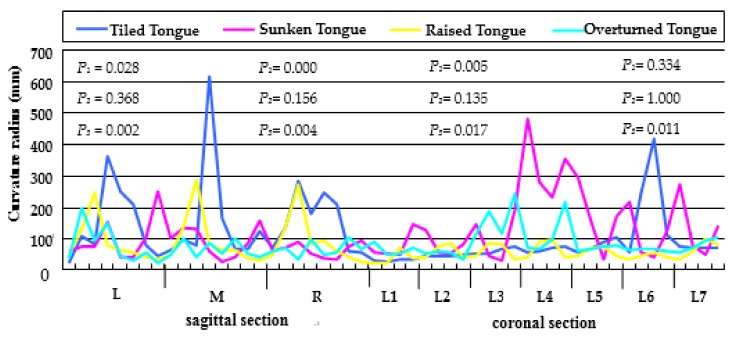
The curvature radius of the feature points in the sagittal sections(L/M/R), coronal sections (L1-L7) within four tongue states and curvature difference statistics. *P* < 0.05 indicates statistical difference, *P* < 0.01 indicates significant statistical difference, *P* < 0.001 indicates extremely significant statistical difference, *P* > 0.05 indicates no significant difference. The different curvature distribution characteristics in each tongue surface characterize their unique properties, resulting in different tongue states.

**Figure 5 sensors-18-04250-f005:**
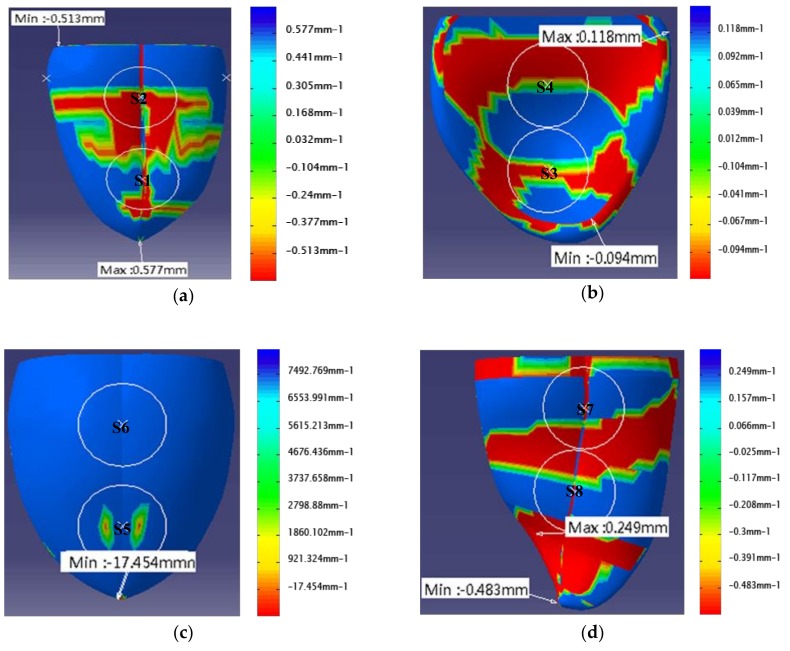
Visualization of Gauss curvature within four tongue states as well as the regional circle of interest selection: (**a**) Tiled Tongue; (**b**) Sunken Tongue; (**c**) Raised Tongue; (**d**) Overturned Tongue.

**Figure 6 sensors-18-04250-f006:**
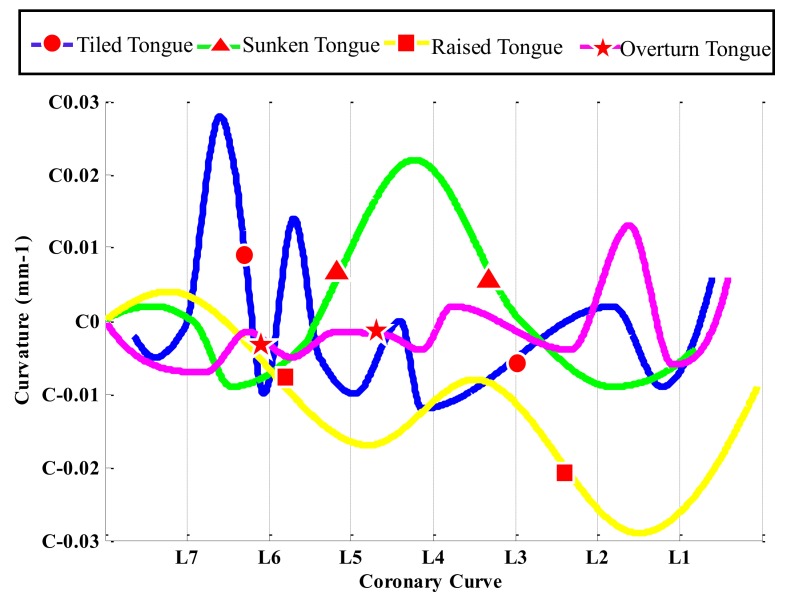
The selected origin of regional circles of interest in the tongue midline for four tongue states. Each curve shown is the curvature curve of the tongue midline of each tongue state. The centers are the middle point of the tongue midline difference between the maximal peak and minimal valley. Each tongue movement state has two origin points.

**Figure 7 sensors-18-04250-f007:**
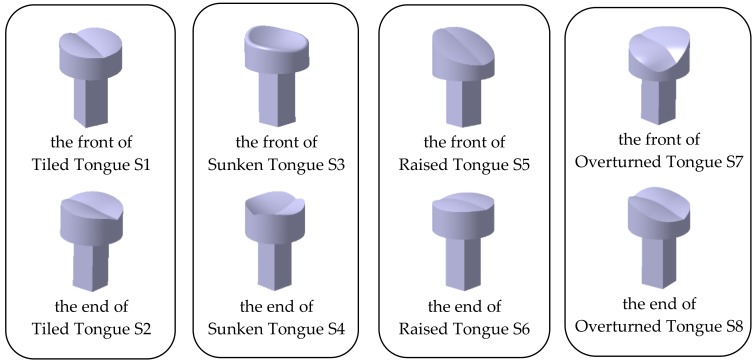
Bionic tongue indenter solid models.

**Figure 8 sensors-18-04250-f008:**
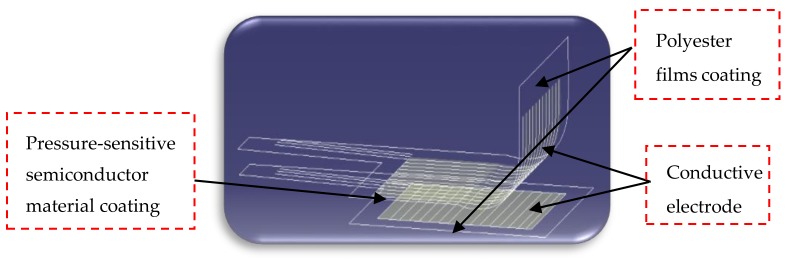
The structural diagram of the arrayed film pressure sensor.

**Figure 9 sensors-18-04250-f009:**
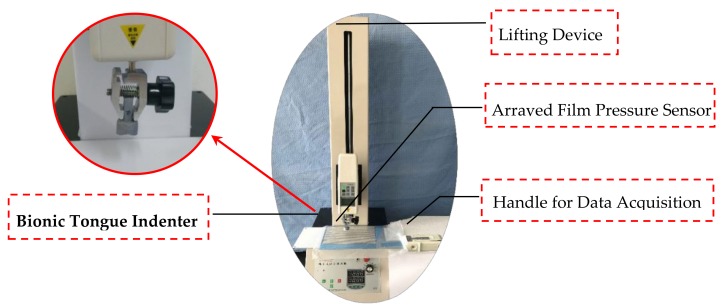
Bionic tongue distributed mechanical testing device. It includes the bionic tongue indenter, the arrayed film pressure sensor, the handle for data acquisition and the lifting device.In order to collect the mechanical information perceived by the arrayed film pressure sensor, which needs to be completed by a data acquisition device, in this paper, we selected a handle for data acquisition produced by the I-Motion Company (Jiangsu, China) to realize the collection of sensing information. When the arrayed film pressure sensor is connected to the handle for data acquisition, the image of the dynamic pressure distribution in the whole sensing area in real time can be displayed and collected by the corresponding computer analysis software. Among them, the function of the handle for data acquisition includes operational amplifier, convertor (ADC) and USB data transmission function. Through the arrayed distribution structure of the ranks sized 52 × 44, the handle for data acquisition can perform multi-channel signal processing and data transmission of up to 2288 points. What’s more, the minimum sampling frequency is 10 ms and the image pressure range is 0–10 MPa.

**Figure 10 sensors-18-04250-f010:**
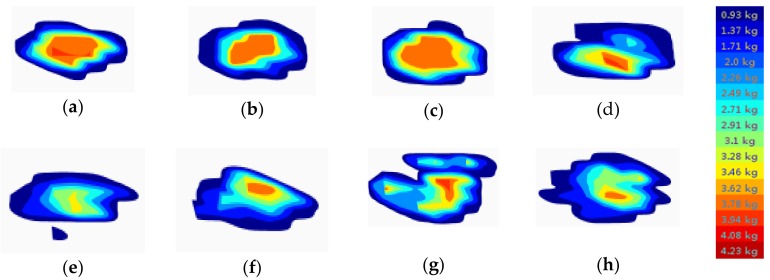
The images corresponding the maximum pressure of the JSL Cookie sample perceived by the eight bionic tongue indenters. The differences in color in the image indicate the difference in pressure data, and a gradual color change from blue to red represents the change in pressure from small to large: (**a**) S1; (**b**) S2; (**c**) S3; (**d**) S4; (**e**) S5; (**f**) S6; (**g**) S7; (**h**) S8.

**Figure 11 sensors-18-04250-f011:**
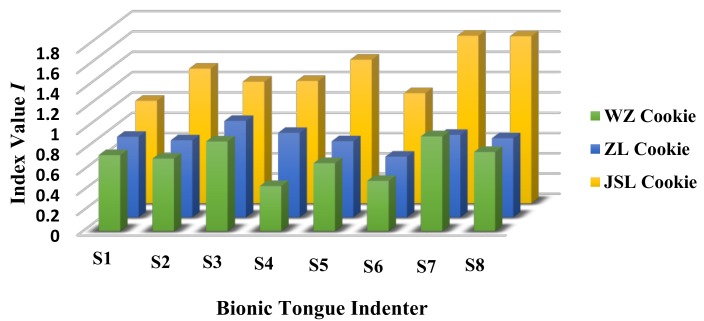
The average of fineness index values of three kinds of cookies calculated by the fineness perception evaluation model based on the mechanical information perceived by eight bionic tongue indenters.

**Figure 12 sensors-18-04250-f012:**
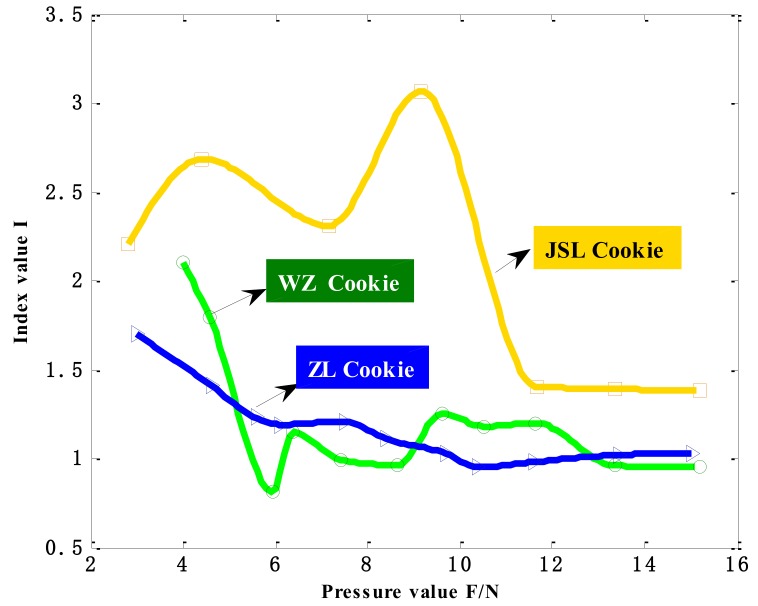
Stress-fineness curves of three kinds of cookies under the bionic tongue indenter S3.

**Figure 13 sensors-18-04250-f013:**
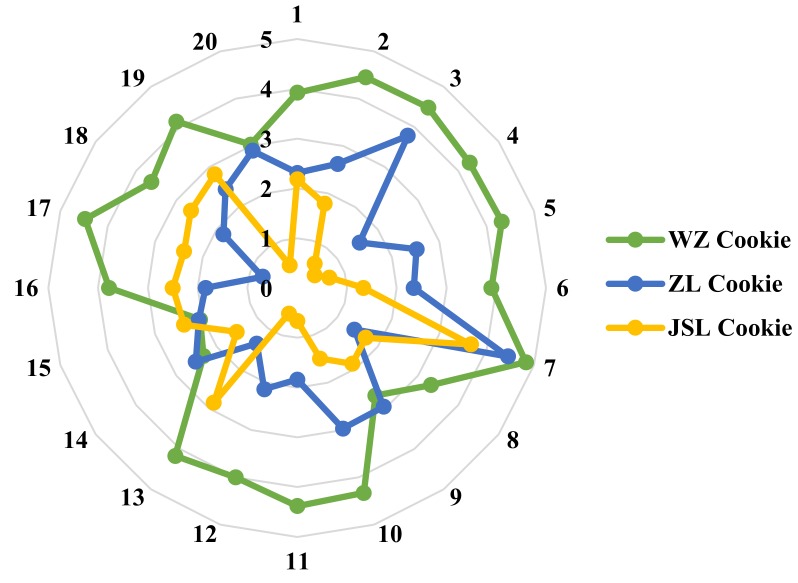
A radar diagram of the different evaluators to the fineness perception of the three kinds of cookies.

**Table 1 sensors-18-04250-t001:** The designed contour parameters of tongue.

	Parameters	Generative Shape Design Module			
		The Width of the Tongue Section *d*/mm	The Height of the Tongue Midpoint *h*/mm	The Round at Both Ends *r_1_*/mm	The Round of the Midline Point *r_2_*/mm
State					
Tiled State		40 ± 11.33	19.5 ± 5.13	3.5	2.5
Sunken State		38.75 ± 15.13	22.75 ± 6.60	/	/
Raised State		31.0 ± 3.16	21.2 ± 3.70	2	/
Overturned State		40 ± 11.33	19.5 ± 5.13	3.5	2.5

Note: The parameter values are presented as the means of different sections (mean ± SD).

**Table 2 sensors-18-04250-t002:** Curvature radius for feature points on the three sagittal section curves (L/M/R).

States	Curvature Radius on Sagittal Sections/mm		
	L	M	R
Tiled Tongue	145.13 ± 118.37 ^a^	160.30 ± 188.24 ^a^	155.13 ± 89.19 ^a^
Sunken Tongue	99.29 ± 71.43 ^b^	92.48 ± 46.74 ^b^	63.67 ± 22.77 ^b^
Raised Tongue	86.13 ± 71.65 ^c^	94.06 ± 85.09 ^c^	94.45 ± 79.51 ^c^
Overturned Tongue	78.91 ± 63.43 ^d^	65.72 ± 26.61 ^d^	67.41 ± 23.90 ^d^

Note: The curvature radius is presented as the mean of eight feature points (mean ± SD). The same letters indicate an insignificant difference in the same line (*P* > 0.05).

**Table 3 sensors-18-04250-t003:** Curvature radius for feature points on seven coronal section curves (L1-7).

States	Curvature Radius on Coronal Sections/mm						
	L1	L2	L3	L4	L5	L6	L7
Tiled Tongue	31.76 ± 3.17 ^a^	45.81 ± 2.02 ^ab^	62.46 ± 10.85 ^ab^	66.43 ± 9.68 ^ab^	80.15 ± 21.54 ^b^	209.41 ± 161.45 ^b^	70.71 ± 2.10 ^ab^
Sunken Tongue	72.63 ± 39.80 ^cd^	79.96 ± 35.08 ^cd^	80.57 ± 52.65 ^c^	337.71 ± 110.35 ^d^	190.41 ± 70.62 ^cd^	109.33 ± 80.60 ^ed^	134.97 ± 100.94 ^cd^
Raised Tongue	38.52 ± 23.53 ^a^	75.92 ± 24.84 ^ac^	76.15 ± 22.33 ^c^	63.93 ± 25.03 ^ac^	58.11 ± 16.70 ^ac^	43.63 ± 9.27 ^ac^	68.36 ± 29.47 ^ac^
Overturned Tongue	65.59 ± 18.75 ^bc^	51.19 ± 11.20 ^b^	165.78 ± 61.53 ^c^	113.63 ± 70.75 ^bc^	69.31 ± 10.48 ^bc^	64.38 ± 3.42 ^bc^	81.06 ± 22.46 ^bc^

Note: The curvature radius is presented as the mean of four feature points (mean ± SD). The same letters indicate insignificant differences in the same line (*P* > 0.05).

**Table 4 sensors-18-04250-t004:** The correlation coefficient between the sensory results and the fineness indexes value of the eight bionic tongue indenters.

Experimental Material	Bionic Tongue Indenter							
	S1	S2	S3	S4	S5	S6	S7	S8
WZ	0.993 **	−0.929	0.793	−0.691	−0.710	0.992 **	−0.994 **	−0.998 **
ZL	−0.913	0.895	−0.986 *	0.981 *	−0.927	0.874	−0.701	0.645
JSL	0.668	−0.890	−0.878	−0.854	−0.952*	−0.997 **	−0.806	0.914

Note: ① * indicates that correlation is significant at the 0.05 level. ② ** indicates that correlation is extremely significant at the 0.01 level.

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
