# Peer review of "Evaluation of Food Fineness by the Bionic Tongue Distributed Mechanical Testing Device"

_sensors, 2018, doi:10.3390/s18124250_

Reviewer 1 Report

Thank You for all the corrections, and your response. I think that the manuscript was improved and that it is suitable for publication in its present form, but as the data for previous samples (banana, apple vs cookie) were also interesting but not representing fineness I would suggest to add them in a supporting information file, as they might be of interest to the readers.

Author Response

Dear Reviewer,

Thank you for the comment concerning our manuscriptEvaluation of Food Fineness by the Bionic Tongue Distributed Mechanical Testing Device (ID: sensors-398940). Your comment gives us great encouragement for my academic writing and scientific effort. Thank you again for your comment. I hope I can learn more knowledge from you. According with your comments and suggestions, we have added the data for previous samples (banana, apple vs cookie) in a supporting information file. Based on your constructive comments and kind suggestions, the modifications and instructions from the whole were answered.

Reviewer 2 Report

The problematic addressed in this study is still relevant. This is a technical paper with standard models and deep simulations.

Nevertheless, this article suffers from lack of experimental measurements and details on the devices used for the collection of experimental data.

The authors have started to introduce the required part in section "2.3. Construction of Distributed Mechanical Testing Device "but Figure 9 is insufficient to understand the test bench used for this experiment.

In my opinion, authors should add more details about the schemes and the principle of the functioning of the devices: lifting device, bionic tongue indenter, arrayed film pressure (the size, the materials used, the biocompatibility aspect of these sensors, etc.). These elements are needed to estimate the sensitivity of the integrated sensors and have more information on the measurement conditions.

Author Response

Dear Reviewer,

 Thank you for the comment concerning our manuscriptEvaluation of Food Fineness by the Bionic Tongue Distributed Mechanical Testing Device (ID: sensors-398940). Your comment gives us great encouragement for my academic writing and scientific effort. Thank you again for your comment. I hope I can learn more knowledge from you. According with your comments and suggestions especially about adding to some details on the devices used for the collection of experimental data, we enriched the details of the device, modified the relevant parts of the manuscript, marked high yellow. And your questions were answered below.

Round  2

Reviewer 2 Report

The authors have addressed all my concerns in this revised version.